# STORMS: A Pilot Feasibility Study for Occupational TeleRehabilitation in Multiple Sclerosis

**DOI:** 10.3390/s24196470

**Published:** 2024-10-07

**Authors:** Lucilla Vestito, Federica Ferraro, Giulia Iaconi, Giulia Genesio, Fabio Bandini, Laura Mori, Carlo Trompetto, Silvana Dellepiane

**Affiliations:** 1Department of Neurosciences, Rehabilitation, Ophthalmology, Genetics, and Maternal and Children’s Sciences (DINOGMI), Università degli Studi di Genova, Largo Paolo Daneo 3, I-16132 Genoa, Italy; lucillavestito79@gmail.com (L.V.); laura.mori@unige.it (L.M.); ctrompetto@neurologia.unige.it (C.T.); 2Ospedale Policlinico San Martino—Istituto di Ricovero e Cura a Carattere Scientifico (IRCCS), Largo Rosanna Benzi 10, I-16132 Genoa, Italy; 3Department of Electrical, Electronics and Telecommunication Engineering and Naval Architecture (DITEN), Università degli Studi di Genova, Via All’Opera Pia 11a, I-16145 Genoa, Italy; 4Struttura Complessa di Neurologia—Ospedale Villa Scassi, Azienda Sanitaria Locale (ASL) 3, Corso Onofrio Scassi 1, I-16149 Genoa, Italy; fabio.bandini@asl3.liguria.it; 5RAISE: Robotics and AI for Socio-Economic Empowerment Ecosystem in Piano Nazionale di Ripresa e Resilienza (PNRR), I-16122 Genoa, Italy

**Keywords:** Internet of Medical Things, home telerehabilitation, multiple sclerosis, remote patient monitoring, occupational rehabilitation

## Abstract

Digital solutions in the field of restorative neurology offer significant assistance, enabling patients to engage in rehabilitation activities remotely. This research introduces ReMoVES, an Internet of Medical Things (IoMT) system delivering telemedicine services specifically tailored for multiple sclerosis rehabilitation, within the overarching framework of the STORMS project. The ReMoVES platform facilitates the provision of a rehabilitative exercise protocol, seamlessly integrated into the Individual Rehabilitation Project, curated by a multidimensional medical team operating remotely. This manuscript delves into the second phase of the STORMS pilot feasibility study, elucidating the technology employed, the outcomes achieved, and the practical, professional, and academic implications. The STORMS initiative, as the genesis of digital telerehabilitation solutions, aims to enhance the quality of life for multiple sclerosis patients.

## 1. Introduction

In addition to physical disabilities, cognitive impairment (CI) affects up to 70% of individuals diagnosed with Multiple Sclerosis (MS). CI typically involves specific deficits in cognitive domains rather than a uniform global decline. Patients with MS (pwMS) may experience difficulties with information processing speed, attention, learning and episodic memory, executive functions, and visuospatial skills. Cognitive impairment can emerge in the early stages of the disease, with about half of individuals reporting minimal or mild cognitive difficulties within the first few years after diagnosis. CI significantly impacts daily life and is a major cause of occupational disability and challenges in activities of daily living (ADL).

Physical rehabilitation and neurorehabilitation are crucial for managing symptoms and enhancing the quality of life (QoL) for pwMS while helping patients maintain independence and functionality. Neurorehabilitation is becoming a valuable therapeutic option for pwMS, and rehabilitative exercise is recognized as an important complementary therapy for MS, with numerous studies demonstrating its significant benefits [1,2], including the promotion of neuroprotective phenomena [3].

Advanced technologies are increasingly being integrated into rehabilitation, ranging from Virtual Reality (VR) and motion capture systems to robot-assisted training, which is especially suited for more complex cases.

Virtual Reality (VR) and exergames have emerged as valuable reinforcement tools in the rehabilitative treatment of individuals with multiple sclerosis [4]. Systematic reviews [5] suggest that VR serves as a motivating and engaging rehabilitation method, potentially enhancing therapeutic compliance [6]. Additionally, by allowing for the selection of different exercises and levels of complexity, VR can adapt to the wide variability in patients’ conditions and disease progression [7].

For individuals with severe disabilities, robotic arms can help in the retraining and recovery of impaired movements. Sophisticated robotic systems, such as exoskeletons, provide precise control over movement, facilitating improvements in mobility and strength. In pwMS, this technology is predominantly utilized for robot-assisted gait training (RAGT). The study in [8] favorably compares low-intensity (8 patients) and high-intensity (8 patients) RAGT against a control group (8 patients). The study in [9] combines RAGT with virtual reality to achieve improved outcomes.

In any case, motion capture is essential in telerehabilitation, providing insights into movement and functional performance [10]. Excluding robotic systems from this discussion, because they inherently perform motion capture, key systems include marker-based and markerless technologies. Marker-based systems involve placing reflective markers on specific anatomical landmarks of the patient’s body. High-speed cameras track these markers to create a 3D model of movements, enabling the analysis of joint angles, gait patterns, and kinematics [11].

Vicon systems are a leading example of marker-based technology, renowned for their accuracy and extensive use in clinical research [12] and rehabilitation for patients with multiple sclerosis (pwMS) [13]. These systems, which typically utilize between 4 and 20 cameras (or even more), provide detailed biomechanical assessments and enable customizable rehabilitation exercises. However, they come with the trade-offs of a complex installation infrastructure and are inappropriate for home telerehabilitation. Qualisys is another prominent example of marker-based technology. Similar to Vicon, Qualisys systems use multiple high-speed cameras to capture detailed movement data, offering high accuracy and extensive use in clinical and research settings [14]. These systems also face similar limitations regarding installation complexity and are less suitable for home-based rehabilitation scenarios.

Markerless systems use high-resolution or depth-sensing cameras combined with advanced algorithms to track movements without the need for physical markers. These systems rely on computer vision techniques to interpret body movements from visual data [15]. They offer advantages such as non-intrusiveness and ease of use, enhancing patient comfort by eliminating physical markers. However, they may be less accurate for complex or occluded movements, and vision-based systems can be affected by lighting conditions and camera angles [16]. Markerless systems are adaptable to various environments and less intrusive.

The Microsoft Kinect Infrared (IR) depth camera is particularly effective for simpler movement tracking, offering a portable and straightforward sensor that does not require complex installation infrastructures. It captures 3D depth information with real-time functional movement analysis and improves patient comfort by eliminating the need for physical markers. Although its accuracy is lower compared to robotic or multi-camera systems, particularly in detecting fine motor details or in complex environments, the Kinect depth camera proves sufficiently accurate for remote motion analysis during rehabilitation [17,18].

The combined use of VR and video games facilitates the development of exergames aimed at promoting both physical and cognitive activity through rehabilitation exercises. The Kinect video game console and sensor capture patient activities without the need for markers or invasive cameras. Compared to other systems like, for instance, the Nintendo Wii [19], Kinect stands out because it does not require controllers or balance boards, making it a more widely used tool in rehabilitation.

The significant role of telerehabilitation for pwMS was demonstrated in recent research by Finkelstein, as described in [20]. The study (16 patients in control group and 29 in intervention group) found that participants in the telerehabilitation group showed improvements in various quality-of-life subdomains, such as physical and emotional roles, pain management, cognitive function, and overall physical and mental health. These improvements were notably greater when compared to a control group that did not participate in the program. Additionally, the telerehabilitation group demonstrated a reduction in urgent care utilization, highlighting the potential long-term benefits of the intervention beyond immediate QoL enhancements.

In our previous work [21], we extensively described the STORMS project [22], funded by Merck through the Digital Innovation Award in Multiple Sclerosis. Based on the use of the ReMoVES system [23,24], it is implemented with the aim of serving as a starting point for the development of digital telerehabilitation solutions to support patients with Multiple Sclerosis, thereby improving their quality of life.

At IRCSS Policlinico San Martino, so far, 34 patients used the ReMoVES system to support their rehabilitation. Twenty MS patients have followed the STORMS project so far.

The main originalities and strengths of this work are:Implementation of an IoMT (Internet of Medical Things) system for the assessment and support of both in-hospital and in-home rehabilitation in people with MS.Utilization and rapid adaptation of the markerless/contactless ReMoVES system in the new target user group.Ability to exercise, monitor, evaluate, and analyze both motor aspects (such as upper limb, lower limb, trunk movement, and balance control) and cognitive aspects (such as attention, memory, working memory, etc.).Provision of a personalized service tailored to the needs of each individual, with an assigned individual care plan.Ease of use, low cost, and integrability with other systems.Robustness and resilience regarding temporary telecommunication problems.Adaptive nonlinear filtering and segmentation of signals for data extraction, analysis, and visualization.

After a brief summary of the exergames extensively described in [21], the study carried out is introduced, and some significant results are provided to demonstrate the ability of the system to observe the patient activity at home and their evolution and eventual progression.

## 2. Materials and Methods

Microsoft [25] proprietary software enables the spatial tracking of the human body and the identification of 25 joints using data from the Kinect depth IR camera, from which skeletal movements are reconstructed and tracked. The most comprehensive study on the error in estimating joint positions is reported in [26], where the authors compare Kinect_v2 with the video-based Qualisys motion capture system. Other studies [27] that compare Kinect with the Vicon system confirm that measurement accuracy is better in the medio-lateral axis than in the vertical and frontal ones. In general, in the context of rehabilitation, the accuracy found is considered acceptable. Signal processing and the exploitation of repeated movements can further improve accuracy. Ref. [28] and others report a comparison between the earlier generation Kinect_v1, Kinect_v2, and Kinect-Azure, showing depth errors in the order of a few millimeters. The more recent Kinect-Azure does not show significant improvements in this regard compared with Kinect v2.

Following a concise overview of the exergames proposed for motor/cognitive rehabilitation activities, the paper proceeds to describe the study design and reports the remotely observed parameters, along with the final evaluation of two patients who completed the study after a period of home-based telerehabilitation.

### Motor/Cognitive Exergames

As described in [21], some new cognitive games have been developed during the STORMS project with the aim of treating some of the most common symptoms of multiple sclerosis such as:Coordination disorders;Balance problems, and dizziness;Vision disturbances, which may also include impaired color vision;Cognitive disorders that incorporate problems with memory and learning;Difficulties in maintaining concentration;Difficulties with attention and computational problems;Inability to perform operations of a certain complexity;Problems in correctly perceiving the environment.

All exergames are based on voluntary limb movements or balance shifts. During execution, the patient requires coordination and performs corrective reactions. Motor and visual coordination is very important for the correct execution of assigned tasks. Direct and indirect measurements of symptoms such as numbness of the body and/or extremities or spasticity that may complicate movement can be obtained when games are played with movement of the pelvis or limbs.

In some exergames, the patient is encouraged in the reaching task by the appearance of consecutive targets on the screen, which must be moved with the movement of the arm. The more targets that are hit, the higher the game score. Such games aim to improve hand–eye coordination and spatial awareness. The Shelf-Cans activity entails placing a colored can on the shelf where similar cans are already positioned. Similarly, in the Owl-Nest game, players are tasked with placing targets of varying difficulty levels into the basket, thus stimulating attention mechanisms.

Other exergames promote trunk balance used to guide an object such as a car or a hot-air balloon and can also be executed when sitting in a wheelchair.

The Owl Nest, Supermarket, Numbers, and Business By Car exercises offer varying levels of difficulty, as detailed in Table 1, ranging from easy to highly challenging. On the other hand, Shelf Cans and Hot Air activities each have a single level, serving as introductory exercises for patients to become accustomed to the system.

In all the exergames, except Business By Car, the total game time is 60 s. Business By Car has a longer duration (90 s) to ensure that the patient can reach the end of the game even if he sometimes makes a mistake along the path. This is in order to define the treatment plan based on the patient’s disability, aimed at selecting the most appropriate game and level to start and continue therapy.

## 3. Study Design

The current pilot study is conducted on a small scale, with the primary purpose of assessing the practicality, feasibility, and potential challenges of the planned clinical study, rather than testing the effectiveness of the intervention [29]. As depicted in Figure 1, several pwMS were recruited at San Martino Hospital at the start time (i.e., time T0), achieving a total of twenty participants included in the study. This sample size fulfills the requirement to estimate the recruitment rate with 90% confidence and a ±15% margin of error, given an estimated rate of approximately 20%.

This step provides a mixed-gender group of adults who vary in age, educational background, and the length of time they have had the disease. We have 20 adults (11 males and 9 females) with the following characteristics:Age range: 35–69 years, mean: 53.8, standard deviation: 7.8,Education range: 8–18 years, mean: 14.6, standard deviation: 3.1,Disease duration range: 8–20 years, mean: 11.6, standard deviation: 3.4,Expanded Disability Status Scale (EDSS) range: 4.5–7, mean: 5.8, standard deviation: 1.0.

After several supervised training sessions at neurorehabilitation clinics (Hospital Study), seven patients were selected to continue telerehabilitation at home under remote medical supervision. Patients were excluded if they had limited familiarity with technology, despite their relatively young age, or if there was no capable caregiver available at home.

Four patients completed the home study for a period of at least one month, up to T2 (3 did not complete due to concomitant clinical issues). The retention rate was 57%.

For the first patients included in the home study, no specific prescription was provided. In the absence of detailed guidance, they were allowed to perform the exercises they found most enjoyable. However, this led to a preference for easier or more enjoyable activities, often at the expense of more challenging or cognitively meaningful ones. As a result, the patients tended to excel in basic exergames rather than advancing their skills through progressively more difficult tasks.

Thanks to this experience, it was decided to take a different approach for the following home studies, prescribing a precise and personalized weekly treatment plan for each patient, with activities distributed from Monday to Thursday.

### 3.1. Inclusion and Exclusion Criteria

For recruitment, the following inclusion criteria were used:Confirmed diagnosis of Multiple Sclerosis (MS).Adults aged between 18 and 60 years.EDSS score ≤ 7.5.Below normal scores on at least two neuropsychological tests.

Exclusion criteria are:Severe mood disorder.Steroid therapy within 2 months prior to the visit.Inability to maintain adequate visual fixation (e.g., nystagmus).Presence of post-chiasmatic perimetric defects.Photosensitive epilepsy.Poor compliance or insufficient motivation to follow the treatment regimen.

The neuropsychological battery consists of tests that are commonly used for cognitive assessment in individuals with disabilities. Specifically, the Brief International Cognitive Assessment for MS (BICAMS) and the Paced Auditory Serial Addition Task (PASAT) at 3 s and 2 s intervals were used. BICAMS includes the Symbol Digit Modalities Test (SDMT), the California Verbal Learning Test II edition (CVLT-II), and the Revised Brief Visuo-Spatial Memory Test (BVMT-R). Patients were recruited if they scored below the 5th percentile for normative data adjusted for age, sex, and education in at least two of the aforementioned tests. Written informed consent was obtained from all participants before the study began.

All cognitive measures were administered at baseline (i.e., time T0), at the end of 10 exergame sessions (T1), and one month after the end of treatment (T2).

In addition to these neuropsychological tests conducted at three fixed moments, daily assessments of motor function were available, thanks to the parameters observed remotely during the execution of the exergames. This continuous monitoring allowed a more detailed and dynamic understanding of the patient’s motor progress. The patient’s perception of effort and fatigue is indirectly assessed through daily observations of key parameters, which will be detailed in the results section. These parameters include the number of exergames played, the number of repeated movements along with their speed and trajectory, game scores, posture, angles, range of motion, etc. Comparing the exercises actually performed with those prescribed is crucial for understanding both the patient’s level of engagement and the appropriateness of the exercise regimen. This approach provides valuable insights into how well the exercises are tailored to the patient’s abilities and needs, helping to optimize the rehabilitation process.

### 3.2. Case Studies

In the following sections of this paper, two patients who used the system for home-based telerehabilitation over an extended period are analyzed in detail. The observed parameters for these patients, referred to as Patient A and Patient B, are described comprehensively.

Patient A is a 49-year-old man, with an EDSS score of 7, who uses a wheelchair. He practiced the ReMoVES system at home for nearly a month without any prescription. The most significant results of his observation are described in Section 4.1.

The patient analyzed in Section 4.2 (Patient B) is a 56-year-old woman (EDSS score of 6, assisted walking) who took the system home for 4 weeks. As a prescription, the medical staff assigned her a weekly schedule of exergames. At the end of each week, a brief report detailing the outcomes of the activities was available to the medical professionals so that the therapy schedule could be improved and personalized for the next week. The results obtained on a weekly basis will be analyzed and the prescriptions will be compared with the actual sessions played by the patient.

Table 2 displays the cognitive assessment scores for Patient A and Patient B at T0, including MMSE, PASAT, SDMT, CVLT-II, and BVMT-R scores along with their respective 5th percentile cutoff values. Both the 3 s and 2 s PASAT were administered. The cut-off takes into account the fact that both patients have less than 12 years of education.

Patient A is below the cutoff in PASAT, SDMT, and BVMT-R; Patient B is below the cutoff in SDMT and BVMT-R. PASAT tests failed (i.e., N).

## 4. Results

### 4.1. First Case Study

Patient A used the ReMoVES system at home for almost a month and his activity was observed throughout the period. Table 3 summarizes the number of sessions played for each exergame, specifying the chosen level. As one can notice, the patient played mainly at the first level of each activity, focusing more on the basic ones (Shelf Cans, Hot Air and Owl Nest).

The large number of HotAir sessions carried out demonstrates the patient’s interest and involvement in what is considered a pleasant activity. The analysis of the acquired data confirms the ease with which the patient is able to carry out this activity in which he almost always reaches maximum performance. In Figure 2, one can see that in most cases, all the targets were caught correctly, apart from a few game sessions where the patient missed a few targets.

Referring to the supermarket exergame, the patient’s cognitive failures concern two types of errors that may occur when taking various objects. A simple error, the so-called “semantic error” occurs when taking a wrong object belonging to the same category as the one described (food or no-food) and is named a “semantic error”; a more serious error occurs when the taken object belongs to a different semantic class than the requested object. From Figure 3, it can be observed that the patient made some errors, even at the semantic level. The errors committed in sessions 1, 2 and 3, respectively, are reported in Table 4 together with the objects in the to-do list. Such errors are most likely the reason why the patient played this game so little and only at the first level. The situation is similar for the exergame Business-by-Car.

The sessions of the Shelf Cans and Owl Nest exercises are much more numerous and will be analyzed in more depth in the following sub-paragraphs.

#### 4.1.1. Shelf Cans Analysis

The learning curve for the Shelf Cans activity reported in Figure 4 depicts the progression of gaming performance with increased experience measured by the number of sessions. Gaming performance is quantified as a percentage increase relative to the first session. As expected, one can observe a low score in the initial sessions and a subsequent steady improvement during the later sessions.

To analyze patient performance in executing the Shelf Cans exergame, various parameters are observed and plotted. Figure 5 provides an overview of the results, illustrating the angles between the optimal trajectories and the trajectories performed by patients, along with the time elapsed in moving the colored cans to their corresponding shelves. The graphs on the left depict the angles between the optimal trajectory and the trajectory executed by the patient. Meanwhile, the graphs on the right illustrate the time taken to complete the required movement. The results for the red, orange, and green trajectories are displayed from top to bottom, respectively.

A noticeable trend is the overall decrease in the time required for each session, indicating an enhanced speed of movement execution over time. Conversely, the angles between the optimal trajectory and the trajectory executed by the patient, serving as an indicator of movement precision, increased for the red can trajectory, while either remaining constant or decreasing for the trajectories of the orange and green cans, respectively.

In addition, a correlation between the angles (indicating precision of movement) and the times (reflecting speed of execution) can be observed. For the red can, there exists a negative correlation (ρ=−0.48), suggesting that as speed increased, accuracy decreased. In contrast, for the orange can, there is a low correlation (ρ=−0.10). Notably, for the green can, a positive correlation is evident (ρ=0.53), indicating the individual’s ability to maintain both speed and precision in their movements.

Regarding the analysis of the patient’s movement, Figure 6 shows the range of motion of the shoulder angles, in the frontal, sagittal and transverse planes, respectively, at each session. We note how, after the first sessions in which the movement was not carried out completely, the values stabilize on measures corresponding to large and correct movements.

Finally, a comparison between Patient A and a group of healthy subjects is depicted in the box plots illustrating the angle between trajectories (see Figure 7). For the red and green trajectories, both the patient’s (in red) and the healthy group’s (in blue) box plots exhibit striking similarity, displaying a certain symmetry in the data distribution. In contrast, for the orange trajectory, the patient’s box plots include some outliers, representing the mistakes made in placing some cans on incorrect shelves.

#### 4.1.2. Owl Nest Analysis

The patient performed 23 sessions of the Owl Nest task, as reported in Figure 8. He played with his right arm, except for one session played with his left (i.e., session 5), without significant differences. He did not play every day of the week, but he did more than one session a day. Match scoring was generally high, except for session 12, which was the last of three on the same day.

Viewing the trajectories of the hand moving across the screen during gameplay provides insight into the accuracy of the movement. Two extreme cases are shown in Figure 9: the trajectories of session 12, in which the patient was visibly tired, and the trajectory of the last day where the patient demonstrated precise control of his movements.

### 4.2. Second Case Study

As described in Section 2, the second study incorporates the prescription provided to the patient undertaking rehabilitation exercises at home. Patient B used the ReMoVES system at home for a 4-week period. Table 5 lists the exergames recommended for each week, along with the level and the minimum number of sessions suggested.

Figure 10 illustrates the patient’s adherence to the prescribed regimen.

Remote observation of the patient’s activity includes an assessment of adherence to the prescription and provides insights into the patient’s condition, including levels of difficulty in task execution, fatigue, and stress. The number of sessions performed each day can be visualized, as shown in Figure 10, with separate plots for each of the four weeks. Sessions conducted on the first day of the week are represented in grey, those on the second day in blue, those on the third day in yellow, and those on the fourth day in orange. The prescribed number of sessions is highlighted with the red bar. The name of non-prescribed activities is written in red.

The graphs depicted in Figure 10 provide comprehensive insights for the medical team and should be interpreted alongside the analysis of execution parameters.

Although the patient did not strictly adhere to the prescription during the first week, she achieved commendable results in the assigned activities. Consequently, for the second week, therapists prescribed more advanced levels of the Owl Nest and introduced the new exergame of Business by Car. During the initial two weeks, the patient frequently engaged in the Owl Nest and Shelf Cans exergames without encountering notable difficulties. Consequently, these games were not prescribed in the subsequent two weeks. Instead, therapists introduced activities in which the patient seemed to face greater challenges, aiming to enhance her concentration and cognitive recovery.

However, it is evident over the course of the weeks that the patient deviated from the prescribed regimen and engaged in numerous advanced activities that were not part of the plan. Additionally, in weeks three and four, she did not participate in the prescribed Numbers exergame. Furthermore, there was a notable decrease in the number of sessions conducted during the final week.

A comparison between actual and prescribed gaming sessions highlighted a significant discrepancy. Various factors likely contributed to this gap, affecting the patient’s participation across different activities and prescriptions. The necessity to juggle multiple activities may have impacted her motivation and willingness to consistently engage in rehabilitation sessions. To enhance future interventions, it may be beneficial to adopt a more comprehensive and balanced approach tailored to individual needs and prescribed activities, which can benefit from the available observations. Addressing factors influencing participation can help ensure greater consistency in exercise adherence and facilitate the achievement of desired outcomes.

#### Business by Car Analysis

In the Business by Car exergame, the player navigates a car along a dynamically generated road. The car responds to lateral trunk movements, turning left or right accordingly. The car’s speed gradually increases until the player veers off the road, incurring a score penalty and resetting the speed. The player must memorize a list of destinations within a specified time frame, ranging from ten to twenty seconds depending on the selected difficulty level. Subsequently, the player drives the car along the path, selecting the correct route at intersections to reach the designated destinations. After completing the errands, multiple-choice questions appear on the screen, pertaining to the list of destinations or details observed in the game scenes or visited buildings. To answer, the player raises their arm and guides an on-screen hand to the answer button. The game offers three difficulty levels:First Level: The player must remember four places to visit. The list of destinations is straightforward without additional information. At intersections, a navigator guides the player to the correct route. If the player takes a wrong turn, a reminder message prompts them to pay closer attention at subsequent intersections. Two questions pertain solely to the list of destinations.Second Level: The player must remember five places to visit. A detailed list of errands appears at the beginning of the game, providing context for the destinations. The navigator does not indicate the correct route, and only a warning message appears if the player deviates from the path. Three questions are posed at the end: two regarding the list of errands and one about a detail of a visited building.Third Level: The player must remember six places to visit. No warning message appears for incorrect turns. At the end, four questions are presented, one related to the initial list of destinations and three concerning details observed in the visited buildings.

The graph displaying correct and incorrect paths per session provides a detailed assessment of the patient’s motor skills during gaming sessions. Correct paths indicate the patient’s ability to accurately follow the game’s predetermined path, while incorrect paths denote deviations or errors made. With this monitoring opportunity, we can assess the patient’s attitude towards exercises and prescriptions; we obtain information on patient involvement and the appropriateness of difficulty levels.

The Business by Car game was prescribed to Patient B 12 times in total, at level 1 for the second and third weeks and at level 2 in the last week. From the analysis of the acquired data, we can understand that the patient was stimulated to play and she totaled 22 sessions against the 12 prescribed.

Some fluctuation in the number of incorrect paths across sessions can be observed which may be indicative of various factors, such as the complexity of the path, level of concentration, or progressive adaptation to challenges. Sessions with a high number of incorrect paths may suggest instances where the patient encountered greater difficulties in following the predefined path, possibly due to increased gameplay complexity. Conversely, sessions with fewer incorrect paths may suggest a greater proficiency in managing the motor challenges posed by the game.

In addition, we can gain much more insight from Figure 11. The patient, after playing level 1 correctly 5 times without errors, tried to play level 3 already in the 8th session, even though it was not prescribed. Since she failed 7 times out of 13 paths, the patient decided to go back to level 1. After two perfectly performed sessions at level 1, she tried level 2 in sessions 11 and 12. Since she encountered some errors, she used level 1 again for a while before switching to the prescribed level 2.

The graph depicted in Figure 12 illustrates the number of correct and incorrect answers per session, providing feedback on the patient’s cognitive abilities. Correct answers indicate the ability to provide accurate responses to questions posed at the end of the game, whereas incorrect answers signify errors or inaccuracies. The trend of correct and incorrect answers offers insight into the evolution of the patient’s cognitive abilities throughout the game program. Sessions with a high number of correct answers may indicate moments of enhanced understanding and concentration, while those with more incorrect answers could suggest instances where the patient struggled with processing information.

In general, we can notice how the cognitive performances in the Business by Car exergame reproduce a situation very similar to the motor performance throughout the 22 sessions played.

## 5. Cognitive Assessement

Table 6 presents the cognitive assessment scores for Patient A and Patient B at T1, at the end of 10 sessions. One can notice a general improvement, which is particularly significant for PASAT-3^″^ and SDMT, which is now above the cutoff for both patients. Finally, one month after the end of treatment (i.e, T2), Table 7 shows that all indexes have been preserved or improved. BVMT-R scores show a significant increase from T0 and after T1 even though they are still below the cutoff but not too far from reaching it.

Finally, Figure 13 and Figure 14 compare all cognitive measures at baseline (T0), at the end of the 10 sessions (T1), and one month after the end of treatment (T2). The general improvement of all the indexes can be observed at once.

## 6. Discussion and Conclusions

In the framework of the STORMS project, the utilization of ReMoVES has been positively embraced by all patients, regardless of age, including elderly individuals typically less inclined towards emerging technologies. This has resulted in increased adherence to rehabilitation protocols, also enhancing the duration of treatment per session. Exergames can be regarded as virtual reality tools, representing an innovative approach to augment motor learning.

The “Challenge Point Hypothesis” [30] suggests the existence of an optimal difficulty level to maintain patient attention without inducing boredom or fatigue, which can lead to frustration and therapy abandonment. Incorporating gamification aspects can enhance treatment outcomes by creating a stimulating and engaging environment.

ReMoVES serves dual purposes as an assessment/measurement outcome tool and a rehabilitation instrument. It enables continuous measurement and monitoring of patient performance, facilitating a detailed functional analysis. This empowers physicians and rehabilitation professionals to identify and/or modify various rehabilitation strategies.

The cognitive tests administered at T0, T1, and T2, as reported in Table 1, Table 6 and Table 7, respectively, demonstrate general improvements in all indices for patients A and B. Notable improvements are observed particularly in the SDMT, BVMT-R, and PASAT scores, although the PASAT-2^″^ and PASAT-3^″^ scores still remain below the cutoff values. Regarding patients’ motor progress, we have proved that continuous monitoring offers a detailed and dynamic understanding of both the patient’s condition and potential improvements. This includes parameters such as the number of exergames played, the frequency and speed of repeated movements, game scores, posture, angles, range of motion, etc. Comparing the exercises actually performed with those prescribed is essential for assessing the patient’s engagement and the suitability of the exercise regimen. This approach yields valuable insights into how well the exercises align with the patient’s abilities and needs, thereby aiding in the optimization of the rehabilitation process. The graphs and figures presented in the results section for both patients indicate a favorable and rapid learning rate, along with general improvements in movement execution, precision, control, and a better understanding of the rehabilitation tasks.

The current literature reveals a general scarcity of publications focused on home-based telerehabilitation, specifically for patients with multiple sclerosis. This gap can be attributed to several factors, including technological limitations and significant challenges related to system acceptability, ease of use, and the need for caregiver support. Many existing studies have primarily addressed rehabilitation technologies within controlled clinical environments or research settings, with less emphasis on their practical application and effectiveness in home-based settings. For instance, the comprehensive review in [5] highlights that only one system specifically addresses home rehabilitation for pwMS. The study published in [31], based on virtual reality (VR), describes a regimen involving two clinic sessions per week and one home session for 32 participants, showing significant differences between the intervention and control groups. Given these challenges, there is a pressing need for further research to explore and address the issues related to the accessibility, usability, and overall effectiveness of home telerehabilitation systems for pwMS. This need underscores the importance and relevance of the current study and its objectives.

The user-friendly interface and cost-effectiveness of ReMoVES provide the opportunity to continue treatment at home, offering advantages in terms of cost and treatment effectiveness.

## Figures and Tables

**Figure 1 sensors-24-06470-f001:**
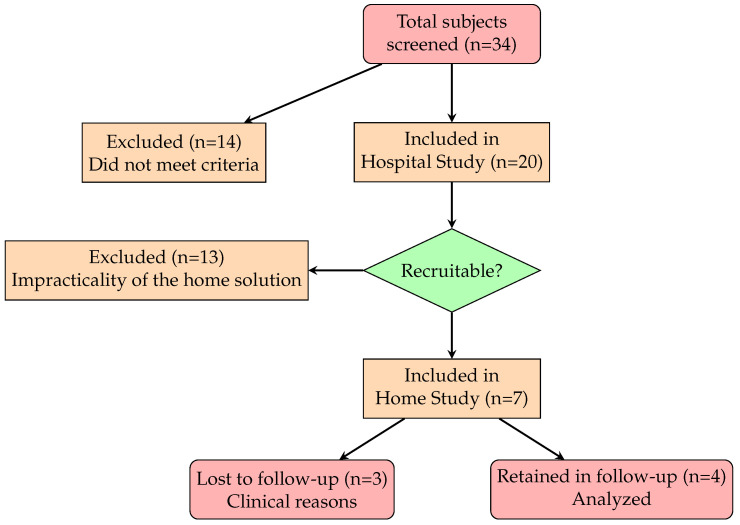
CONSORT flow diagram of the study.

**Figure 2 sensors-24-06470-f002:**
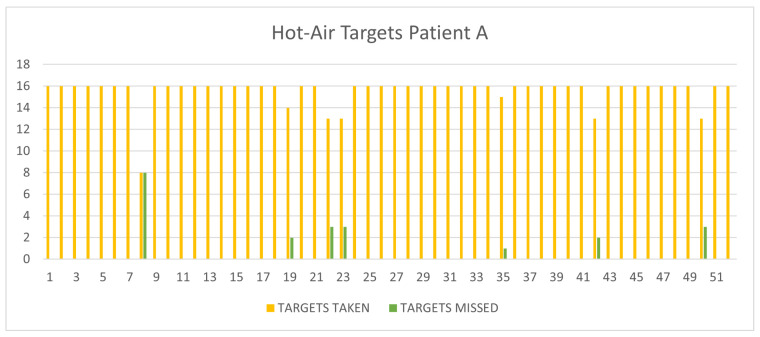
Targets taken and targets missed in the 52 HotAir sessions (Patient A). The x-axis represents the number of sessions played, while on the y-axis, we find the number of targets taken (yellow) and target missed (green).

**Figure 3 sensors-24-06470-f003:**
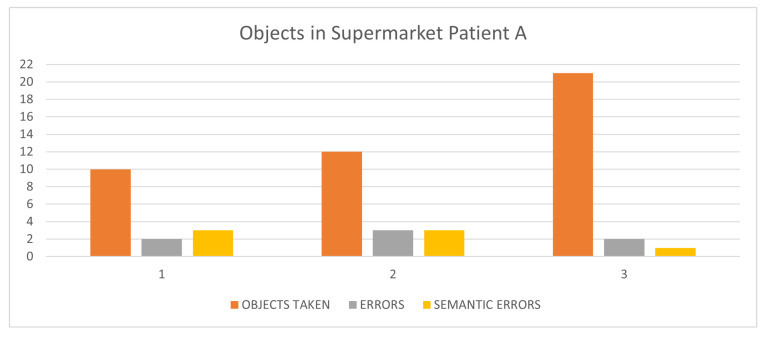
Objects taken, errors and semantic errors in supermarket sessions (Patient A). Semantic errors (in yellow) are considered less “serious” than errors (in grey).

**Figure 4 sensors-24-06470-f004:**
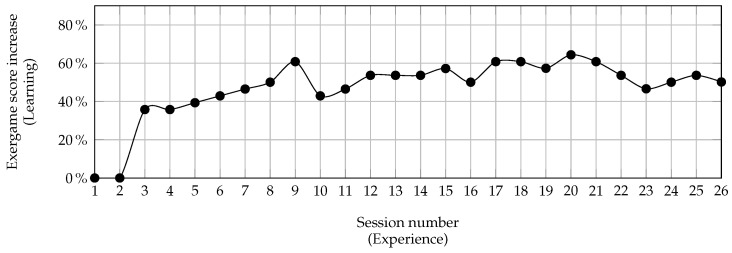
Patient A: the learning curve for the Shelf Cans over 26 sessions.

**Figure 5 sensors-24-06470-f005:**
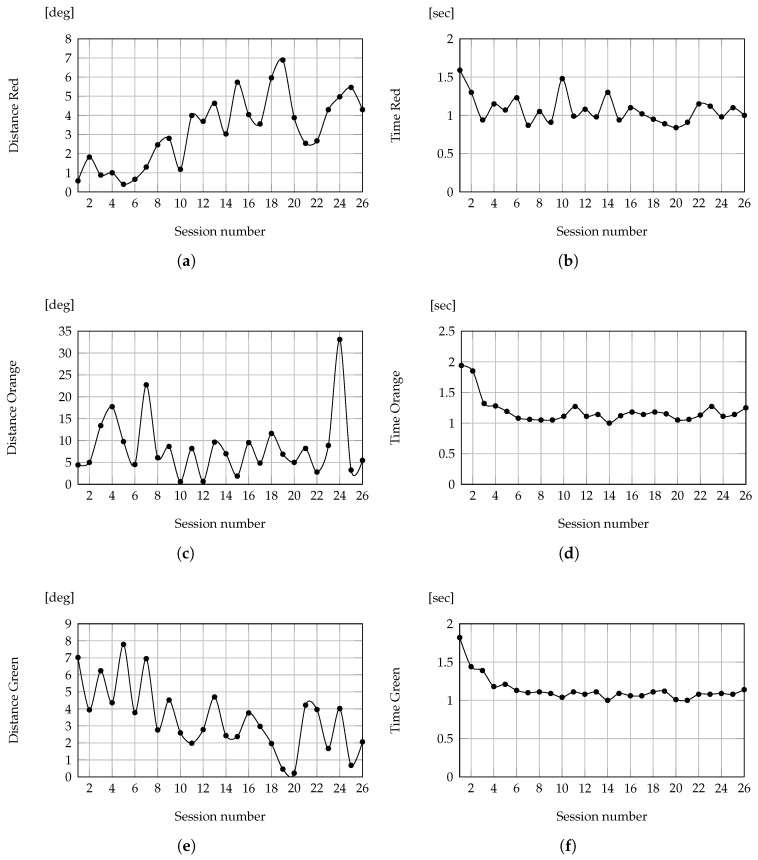
Angles between the optimal trajectory and the one performed by the patient (**left**). Execution times to perform the required movement (**right**). Subfigures (**a**,**b**) refer to the red cans, (**c**,**d**) to the orange cans, and (**e**,**f**) to the green cans.

**Figure 6 sensors-24-06470-f006:**
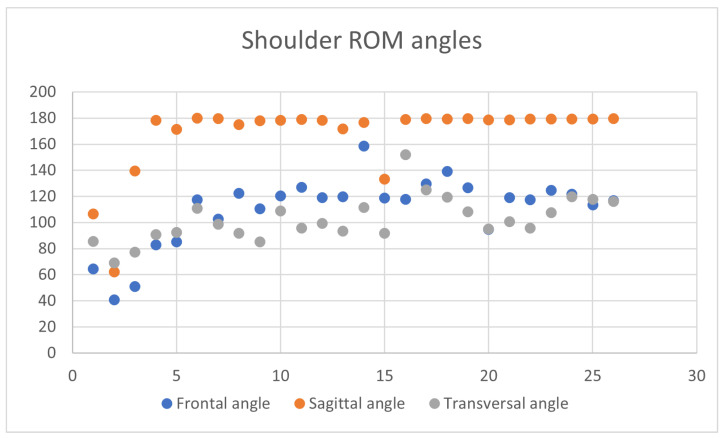
Shoulder angle range of motion: Patient A.

**Figure 7 sensors-24-06470-f007:**
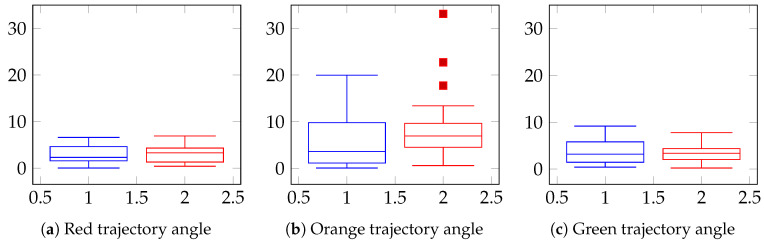
Box plot depicting the values of trajectory angles of Healthy Subjects (HS, blue) and Patient A (patA, red).

**Figure 8 sensors-24-06470-f008:**
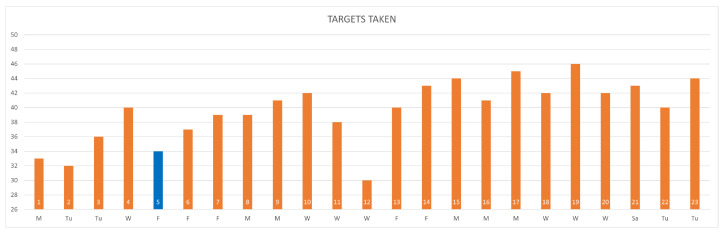
Targets taken at each session, Owl Nest, Patient A. Orange sessions have been played with right arm; the blue session with the left arm.

**Figure 9 sensors-24-06470-f009:**
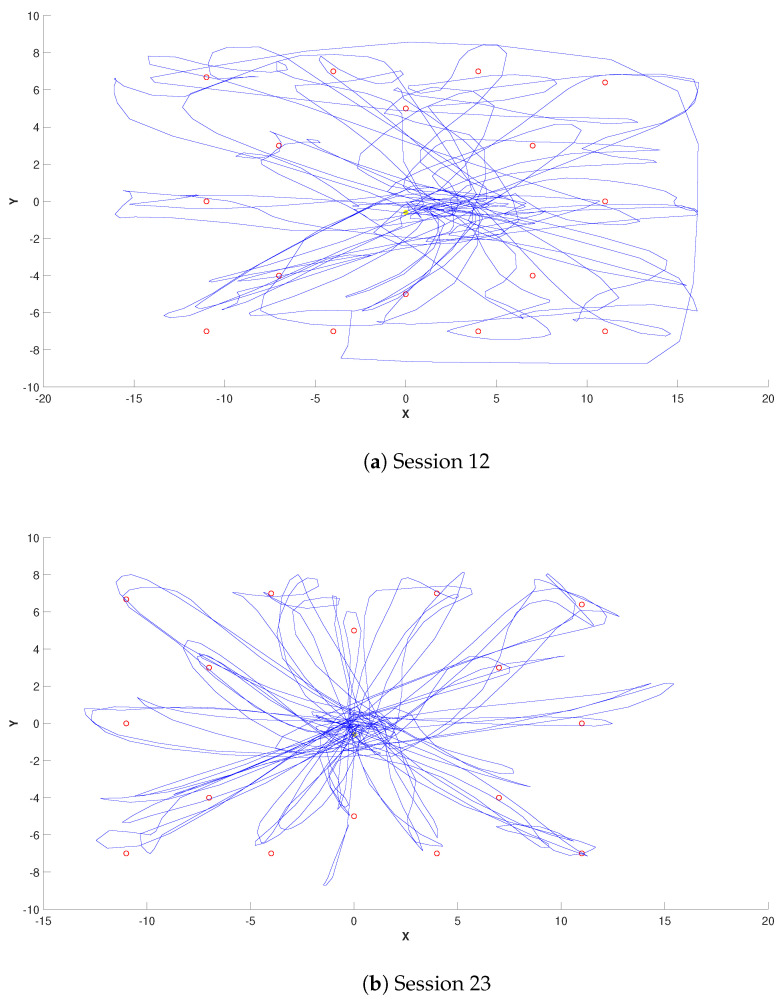
Trajectories during session 12 (**a**) and session 23 (**b**) of Owl Nest activity of Patient A.

**Figure 10 sensors-24-06470-f010:**
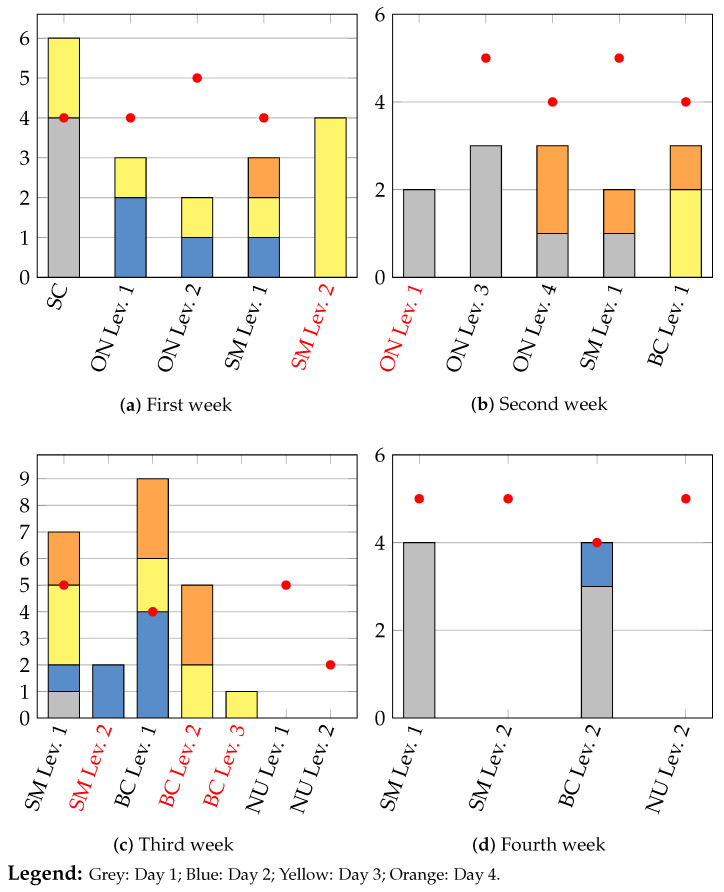
Number of sessions performed each week. The x-axis shows the exercises. The exercises in red highlight the execution of non-prescribed activities, while the red circles represent the doctor’s prescription per exercise.

**Figure 11 sensors-24-06470-f011:**
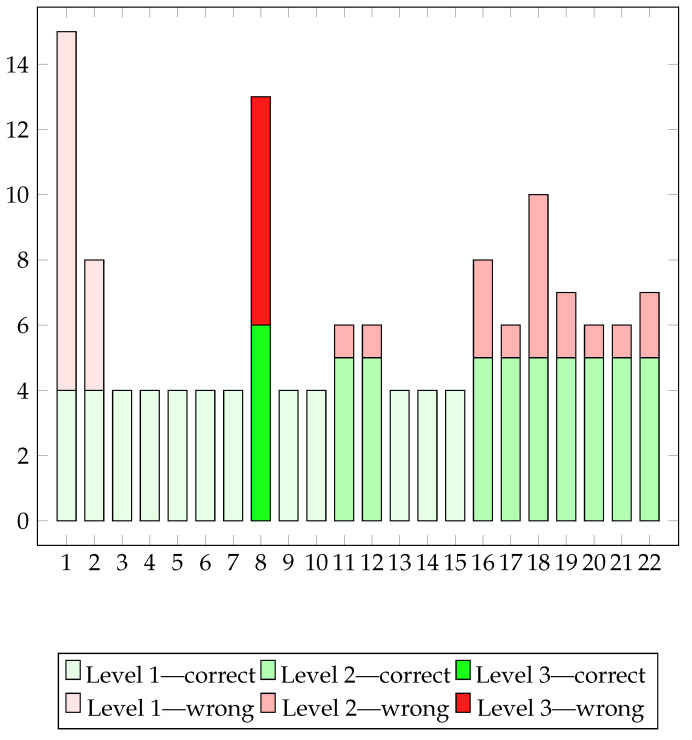
Number of correct and incorrect paths per session.

**Figure 12 sensors-24-06470-f012:**
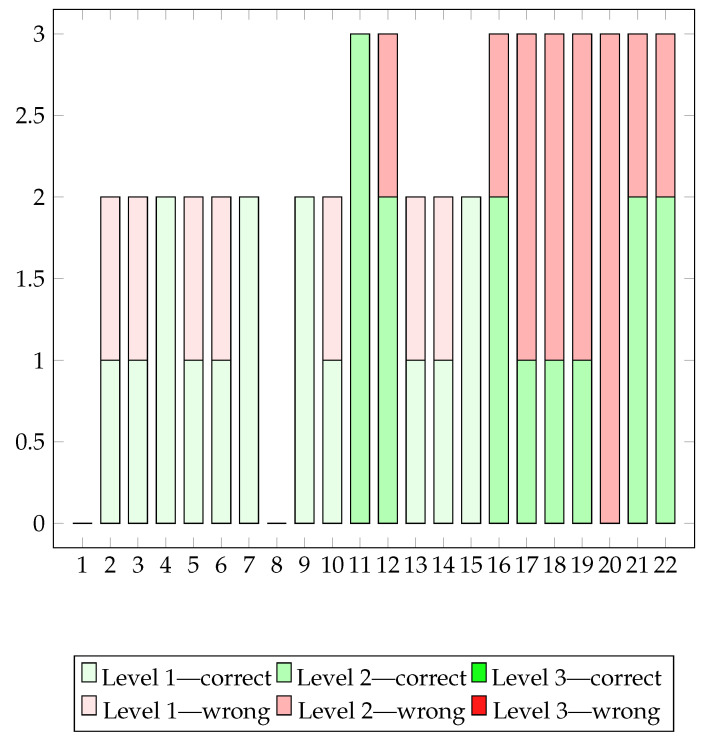
Number of correct and incorrect answers per session.

**Figure 13 sensors-24-06470-f013:**
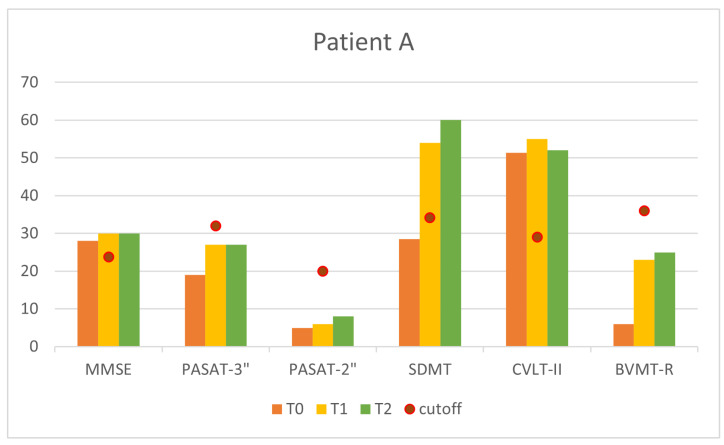
Cognitive measures at T0,T1, and T2 for Patient A.

**Figure 14 sensors-24-06470-f014:**
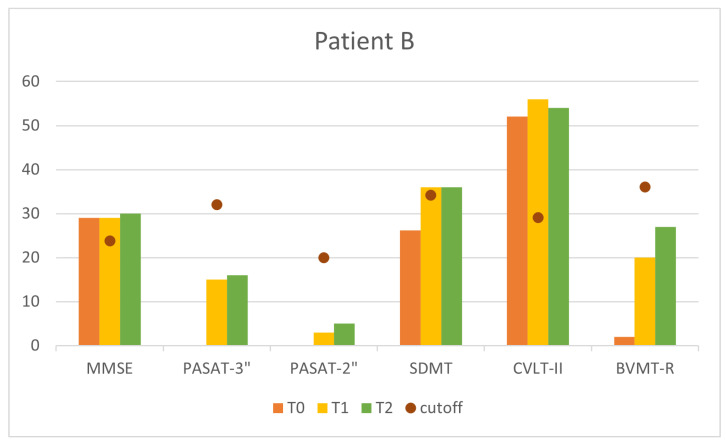
Cognitive measures at T0,T1, and T2 for Patient B.

**Table 1 sensors-24-06470-t001:** Exergames, available levels, required movement, and cognitive task.

Exergame	Levels Number	Voluntary Movement	Cognitive Task
Shelf Cans	1	arm	attention
Owl Nest	4	arm	attention
Numbers	5	Limb	attention
Supermarket	4	arm	attention&memory
Business By Car	3	balance	attention&memory
Hot Air	1	balance	attention

**Table 2 sensors-24-06470-t002:** Cognitive assessment scores for Patient A and Patient B at T0. Below-grade values are highlighted in red. The 5th percentile cutoff values.

	MMSE		PASAT-3^″^		PASAT-2^″^		SDMT		CVLT-II		BVMT-R
	**Score**	**Cutoff**	**Score**	**Cutoff**	**Score**	**Cutoff**	**Score**	**Cutoff**	**Score**	**Cutoff**	
Patient A	28.62	23.8	19	32	5	20	28.5	34.20	51.32	29.05	6/36
Patient B	29	23.8	N	32	N	20	26.2	34.20	52.05	29.05	2/36

**Table 3 sensors-24-06470-t003:** Number of sessions performed by Patient A for each exergame.

Exergame	Level	# Session
Shelf Cans	1	26
Owl Nest	1	23
Numbers	1–2	4
Supermarket	1	3
Business By Car	1	3
Hot Air	1	52

**Table 4 sensors-24-06470-t004:** Error objects correspond to those highlighted in grey in Figure 3.

Session	To-Do List	Wrong Objects
1	Banana–Pretzel–Watermelon	Book–Book
2	Chicken–Watermelon–Mushrooms	Glasses–Cap–Book
3	Pretzel–Mushrooms–Banana	Book–Book

**Table 5 sensors-24-06470-t005:** Plan of care of the activities prescribed for 4 weeks.

Week	Exergame	Level	Prescription
1	Shelf Cans	-	4
	Owl Nest	1	4
	Owl Nest	2	5
	Supermarket	1	4
2	Owl Nest	3	5
	Owl Nest	4	4
	Supermarket	1	5
	Business By Car	1	4
3	Supermarket	1	5
	Business By Car	1	4
	Numbers	1	5
	Numbers	2	2
4	Supermarket	1	5
	Supermarket	2	5
	Business By Car	2	4
	Numbers	2	5

**Table 6 sensors-24-06470-t006:** Cognitive assessment scores for Patient A and Patient B at T1. Values below the cutoff are highlighted in red. Values in blue are below the cutoff but not far from reaching it.

	MMSE		PASAT-3^″^		PASAT-2^″^		SDMT		CVLT-II		BVMT-R
	**Score**	**Cutoff**	**Score**	**Cutoff**	**Score**	**Cutoff**	**Score**	**Cutoff**	**Score**	**Cutoff**	
Patient A	30	23.8	27	32	6	20	54	34.20	55	29.05	23/36
Patient B	29	23.8	15	32	3	20	36	34.20	56	29.05	20/36

**Table 7 sensors-24-06470-t007:** Cognitive assessment scores for Patient A and Patient B at T2.Values below the cutoff are highlighted in red. Values in blue are below the cutoff but not far from reaching it.

	MMSE		PASAT-3^″^		PASAT-2^″^		SDMT		CVLT-II		BVMT-R
	**Score**	**Cutoff**	**Score**	**Cutoff**	**Score**	**Cutoff**	**Score**	**Cutoff**	**Score**	**Cutoff**	
Patient A	30	23.8	27	32	8	20	60	34.20	52	29.05	25/36
Patient B	30	23.8	16	32	5	20	36	34.20	54	29.05	27/36

## Data Availability

Data are contained within the article.

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
