# Peer review of "STORMS: A Pilot Feasibility Study for Occupational TeleRehabilitation in Multiple Sclerosis"

_sensors, 2024, doi:10.3390/s24196470_

Round 1

Reviewer 1 Report

Comments and Suggestions for Authors

The topic of study is relevant and novel, however, I have some doubts regarding the methodology, and I think they should be clarified so that other authors have a good understanding of the protocol carried out when reading their manuscript. In addition, I make some constructive comments to improve the quality of the study.

-          I recommend that authors put the type of study carried out in the title and the methodology section so that authors who want to read it do not have doubts about it.

-          I seem to understand from reading the article that 20 patients were included at the beginning of the study, however, they only analyzed the results of 2 patients (patient A and patient B). What is it due to? Results from a single patient cannot be extrapolated to the entire multiple sclerosis population. They must calculate the sample size necessary for proper external validity of the study and describe the results in a significant sample.

-          I recommend including a flow chart of the study patients

-          Were the patients assigned to protocols A and B randomly?

-          Patient A is 49 years old and patient B is 56, could this be a significant age difference influencing the results?

-          The authors must include the statistical significance values ​​in Tables 6 and 7.

-          The authors comment that one of the main strengths of this study is its ability to exercise, monitor, evaluate, and analyze the functional capacity of patients; however, they have not analyzed this variable, they are only limited to analyzing cognitive aspects.

-          Did they measure the effort perceived by the patients when performing the exercise?

-          Were the evaluators blind? must be described.

-          The discussion should be improved by comparing the results obtained in this study with the results of similar studies carried out by other authors.

A cordial greeting

Author Response

Please see the attachment. A pdf file of the manuscript with changes highlited in red is also enclosed in the Supplementary Files.

Reviewer 2 Report

Comments and Suggestions for Authors

1.      In the introduction, the authors mention the use of markerless sensors. The reviewer suggests that the authors provide specific examples of both marker-based and markerless sensors and conduct a detailed comparison of their advantages and disadvantages to better highlight the novelty of this work.

2.      Regarding patient recruitment, the reviewer inquires whether a statistical power analysis was conducted to determine the appropriate sample size.

3.      From a hardware perspective, the reviewer asks if there are any new innovations in this work compared to the authors’ previous studies.

4.      Given that this system is intended for rehabilitation, the reviewer advises the authors to present quantitative data demonstrating the improvement in patients with multiple sclerosis or cognitive impairment after long-term use of the system.

5.      The manuscript primarily discusses individual patients. The reviewer notes a lack of a statistical summary of all patients involved in the study, which is necessary to substantiate the system's effectiveness.

Author Response

Please see the attachment. A PDF file of the manuscript with changes highlighted in red is also included in the Supplementary Files.

Round 2

Reviewer 1 Report

Comments and Suggestions for Authors

The authors have made the requested improvements in the manuscript, improving its quality and clarity in the explanation of the study carried out. However, they should improve regarding the flowchart, citing the figure at the bottom, and following the CONSORT flow diagram model where the number of subjects included in the study, number of subjects excluded, number of losses. during follow-up and reason, and finally number of subjects analyzed.

Author Response

Thank you for the interesting suggestion. The previous flowchart has been replaced by the CONSORT flow diagram, which now reports the number of subjects involved at each phase. In addition, the following  sentence has been added to the paragraph explaining the reason for exclusion from the Home Study.

"After several supervised training sessions at neurorehabilitation clinics (Hospital Study), 
seven patients were selected to continue telerehabilitation at home under remote medical 
supervision. Patients were excluded if they had limited familiarity with technology, despite 
their relatively young age, or if there was no capable caregiver available at home. "

The new paper version is in the file enclosed.

Best Regards, Silvana Dellepiane
